# Actively Inferring Optimal Measurement Sequences

## Abstract

Measurement of a physical quantity such as light intensity is an integral part of many reconstruction and decision scenarios but can be costly in terms of acquisition time, invasion of or damage to the environment and storage. Data minimisation and compliance with data protection laws is also an important consideration. Where there are a range of measurements that can be made, some may be more informative and compliant with the overall measurement objective than others. We develop an active sequential inference algorithm that uses the low dimensional representational latent space from a variational autoencoder (VAE) to choose which measurement to make next. Our aim is to recover high dimensional data by making as few measurements as possible. We adapt the VAE encoder to map partial data measurements on to the latent space of the complete data. The algorithm draws samples from this latent space and uses the VAE decoder to generate data conditional on the partial measurements. Estimated measurements are made on the generated data and fed back through the partial VAE encoder to the latent space where they can be evaluated prior to making a measurement. Starting from no measurements and a normal prior on the latent space, we consider alternative strategies for choosing the next measurement and updating the predictive posterior prior for the next step. The algorithm is illustrated using the Fashion MNIST dataset and a novel convolutional Hadamard pattern measurement basis. We see that useful patterns are chosen within 10 steps, leading to the convergence of the guiding generative images. Compared with using stochastic variational inference to infer the parameters of the posterior distribution for each generated data point individually, the partial VAE framework can efficiently process batches of generated data and obtains superior results with minimal measurements.

## 1    Introduction and overview

In many circumstances, data collection incurs a cost. This cost may be in terms of acquisition time, invasion of or damage to the environment and storage. Identifying which, out of a range of data measurements, to collect next is potentially a valuable cost saving activity. Importantly, it also facilitates fast decision making (Horvitz & Barry, 1995). Data minimisation is an important consideration for compliance with data protection laws worldwide [1]. In defense situations, the requirement for covert human intelligence means that the act of taking measurements can also pose a security risk. [2]

In this work, our overall aim is to identify an optimal measurement sequence. We develop an active sequential inference algorithm that uses a low dimensional data representation to infer its high dimensional state conditional on partial measurement of that state. Reducing dimension allows for more efficient exploration of the space of state possibilities. The state and types of tasks we are primarily interested in are image/scene reconstruction and related classification tasks using photon based imaging and sensing technology such as a single pixel camera (Higham et al., 2018). However, the method is applicable to a broader range of activities.

The overall aim is to customise a generative probabilistic model to provide the agent or user with different reconstruction scenarios conditional on partial measurements. Given a suitably large dataset of task relevant

---

[1] https://ico.org.uk/for-organisations/uk-gdpr-guidance-and-resources/
[2] https://www.gov.uk/government/publications/covert-human-intelligence-sources-code-of-practice-2022/covert-human-intelligence-sources-revised-code-of-practice-accessible

data, an appropriate generative model is the variational autoencoder (VAE) (Kingma & Welling, 2014). A VAE learns to encode data in the convenient form of a low dimensional multivariate Gaussian and to decode this representation back to data space. This provides a means to obtain different reconstructions through manipulation of a low dimensional latent space. This neural network model is trained on relevant data to learn the underlying distribution of the training data and used to generate new data. A VAE comprises an encoder (a map from data space to a low dimensional latent representative space) and a decoder (a map from the low dimensional space back to data space). To achieve this the VAE introduces latent variables and the objective of a VAE is to understand the true posterior distribution of the latent variables. A VAE accomplishes this by employing an encoder network to approximate the genuine posterior distribution with a learned approximation. Once trained, samples from the prior on the latent space can be pushed through the decoder in order to obtain new generated data. Similarly, samples from the posterior can be pushed through the decoder in order to obtain data conditional on the input data. Here we adapt the encoder to map partially measured data to the latent space. This necessitates creating a training set of partially measured data. Once trained, the partial encoder and the original decoder are used by the algorithm to sequentially reason about the full state and choose the next measurement. The probabilistic model we are inferring over is capturing both the (approximate) data-generating process and the noisy measurement model.

*The main idea of variational methods is to cast inference as an optimization problem (Blei et al., 2017).* Stochastic variational inference (SVI) (Hoffman et al., 2013) can be used to infer the posterior probability distribution for specific data and a given set of measurements but requires multiple computation steps and thus is prohibitively slow when required to estimate many possible measurement sets. We propose a hybrid approach, in a similar spirit to Kim et al. (2018) but novel in our context, to combine the strengths of VAE and SVI. We use the partial encoder to choose between patterns and integrate robust SVI when a pattern has been selected for inferring the posterior distribution parameters based on actual measurements. The algorithm is developed to actively select the next best measurement. It starts by pushing samples from the prior distribution on the latent space through the decoder to obtain candidate images. Possible measurements on these candidate images are estimated, forming an indexed set. The problem can be thought of as choosing the next measurement index from a set of possible measurement indexes. The partial encoder or SVI is used to characterise the posterior distribution and hence provide a score under that distribution for each possible measurement index for each candidate image. There are different ways to define this score. We consider two approaches. First, we choose the index with the highest average (over the candidate images) likelihood score. Second, we choose the index with the least uncertainty score. The aim is to develop a method that is flexible to the task context and measurement basis.

## 1.1 Related work

There is a large literature on data driven models for solving inverse problems (Arridge et al., 2019). In particular, generative variational models have been developed for a range of inverse problems in imaging (Habring & Holler, 2022). These include inpainting, denoising, deblurring, super resolution and JPEG decompression. Here, we focus on acquiring data rather than how to utilise data once collected. *Quantitative methods for optimizing data acquisition have been explored in statistics and machine learning literature (Rainforth et al., 2024). Bayesian experimental design is a powerful model-based framework for choosing designs optimally using information-theoretic principles. This includes adaptive design Bayesian active learning (MacKay, 1992a), sequential (Foster et al., 2021) and Bayesian optimization (Mockus, 1989; Garnett, 2023). Other related topics include active feature acquisition (Saar-Tsechansky et al., 2009), active multi-modal acquisition (Kossen et al., 2023), active perception (Bajcsy et al., 2018), Bayesian active learning (MacKay, 1992b), active reinforcement learning (Andreopoulos & Tsotsos, 2013) and active vision (Whitehead & Ballard, 1990).*

A generative model-based approach to Bayesian inverse problems, such as image reconstruction from noisy and incomplete images, is developed in Böhm et al. (2019). Their inference framework makes use of a VAE to provide complex, data-driven priors that comprise all available information about the uncorrupted data distribution and enables computationally tractable uncertainty quantification in the form of posterior analysis in latent and data space. Our approach differs in how we extend the VAE to the data. Our focus is identifying high value data points so we propose a different VAE to provide the prior and facilitate posterior analysis in latent and data space. Similarly our approach can be adapted to different data systems without

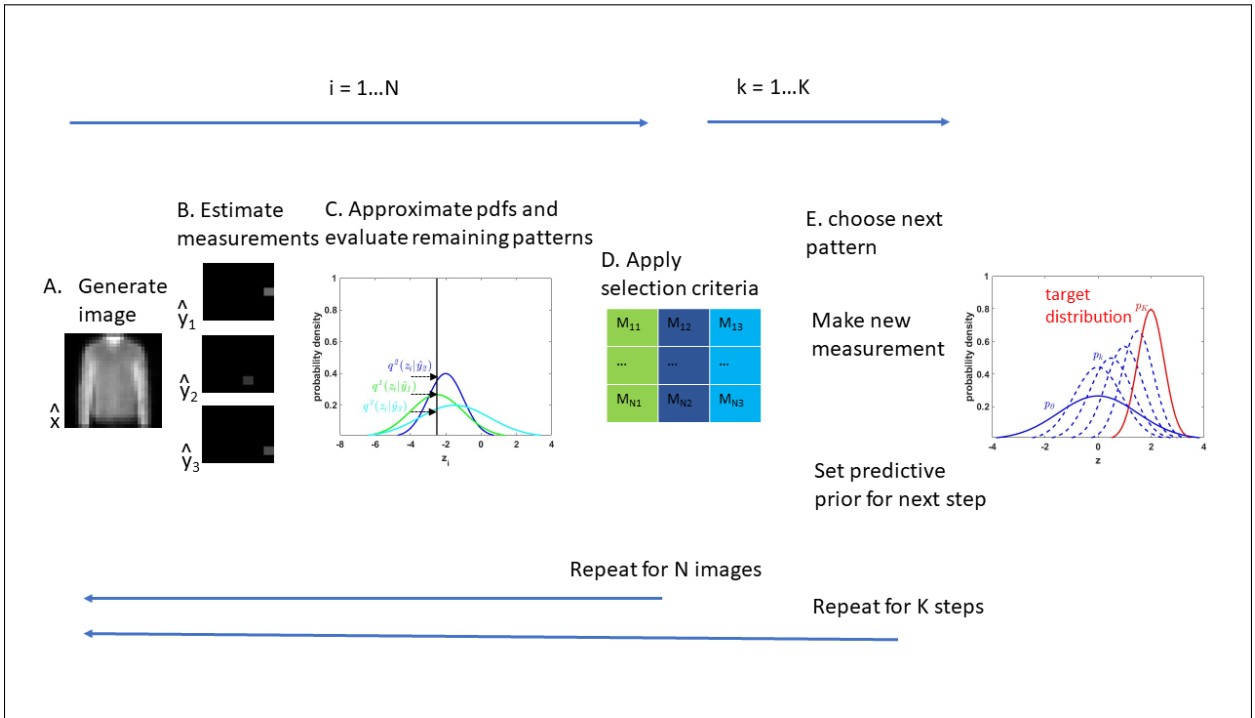

Figure 1: Active Learning. At step $k = 0$, for each of $N$ starting points, sample $\mathbf{z}_i$ from the prior for step $k = 0$, push $\mathbf{z}_i$ through the decoder to obtain a generated image $\hat{\mathbf{x}}_i$ and estimate possible measurements $\hat{\mathbf{y}}_i$ for each pattern not yet measured (e.g. $\hat{\mathbf{y}}_1$, $\hat{\mathbf{y}}_2$ and $\hat{\mathbf{y}}_3$ illustrated above). Use the partial encoder to approximate the posterior with probability distributions and estimate probability densities $q^1(\mathbf{z}_1|\hat{\mathbf{y}}_1)$, $q^2(\mathbf{z}_1|\hat{\mathbf{y}}_2)$ and $q^3(\mathbf{z}_1|\hat{\mathbf{y}}_3)$ (illustrated above). Select the pattern which maximises the chosen expression and take the measurement associated with this pattern. Update the measurement set and update the predictive prior for the next step. By repeating these steps, we move towards the target distribution $p(\mathbf{z}|\mathbf{x})$.

retraining the core VAE. Our extended VAE can be used repeatedly for the sequence of measurements whereas the method in Böhm et al. (2019) has as its focus corrupted or missing data and is designed to tackle recovery of data for a given instance rather than exploring uncertainty in a sequential manner.

A partial variational autoencoder (Partial VAE) is introduced in Ma et al. (2019b) to predict problem specific missing data entries given a subset of of the observed ones. The model is combined with an acquisition function that maximises expected information gain on a set of target variables. The VAE based framework is extended to a Bayesian treatment of the weights in Ma et al. (2019a). Our work overlaps in that we develop a VAE and use it to identify high value data points. However it differs in that we are concerned with measurements of data taken with respect to a basis. We adapt our encoder to the measurement basis rather than the problem and hence extend the active learning application to image reconstruction using sensing technology. The workshop paper by Saar-Tsechansky et al. (2009) is also relevant to our work and extends the work (Ma et al., 2019b) by adding transformer components to the partial VAE architecture. One advantage of our partial encoder, over these works, is that given a full VAE on a domain, the partial encoder can be trained on different measurement basis.

In Section 2 we describe our method for posterior inference given measurements. The active sequential measurement algorithm is outlined in Section 3. An overview of the algorithm is provided in Figure 1.

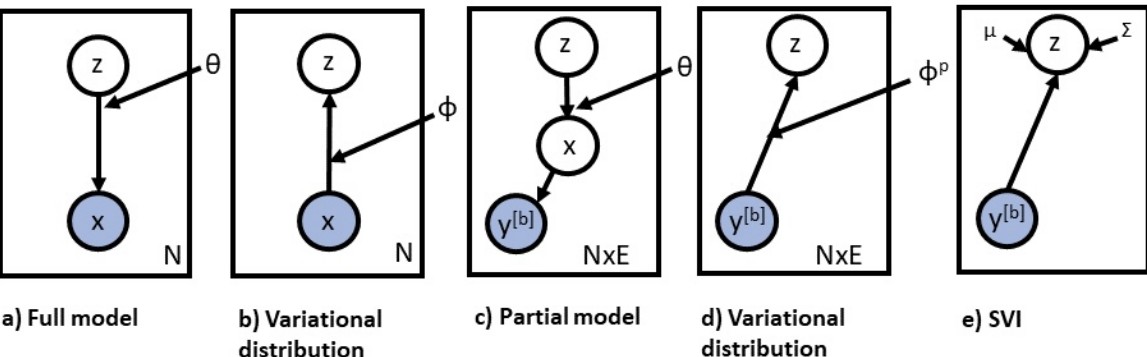

Figure 2: Graphical representation of the VAE model. An image $\mathbf{x}$ is generated by a random variable $\mathbf{z}$ parameterized by a deep neural network with parameters $\theta$, a). A variational distribution parameterized by a deep neural network with parameters $\phi$ is introduced to infer $\mathbf{z}$ given $\mathbf{x}$, b). The encoder and decoder are trained together using $N$ images. To train the partial encoder we simulate $N \times E$ measurements $\mathbf{y}^{[b]}$ where $b$ is a subset of patterns, c). The partial encoder parameterized by $\phi^p$ is trained with the original decoder, d). The SVI method involves inferring the mean $\mu$ and variance $\Sigma$ for each image individually, e).

## 2 Method for posterior inference given measurements

We describe the underlying VAE, Section 2.1, the extension of this framework to sensor measurements, Section 2.2, and the SVI method, Section 2.3. We then introduce the partial encoder, Section 2.4, and training of the partial encoder, Section 2.5.

### 2.1 Underlying VAE model

We assume that an image, $\mathbf{x}$, is generated by a latent random variable, $\mathbf{z}$, in a complex non-linear manner, parameterized by a deep neural network decoder, $D_\theta$, with parameters $\theta$. The structure of this model is represented graphically in column (a) of Figure 2. To do inference in this model we introduce a variational distribution to approximate the posterior distribution of $\mathbf{z}$ given $\mathbf{x}$. Through training the VAE learns a function that maps each $\mathbf{x}$ to the parameters of posterior densities. Typically this mapping encodes the mean, $\mu$, and variance, $\boldsymbol{\Sigma}$, of a Gaussian distribution, $\mathcal{N}(\mu, \boldsymbol{\Sigma})$, in latent space. This function is also parameterized by a deep neural network with parameters $\phi$ (encoder network $E_\phi$) and the variational distribution, $q_\phi(\mathbf{z}|\mathbf{x})$, is represented graphically in column (b) of Figure 2. The goal of training is to find optimal values for $\theta$ and $\phi$ so that the model, $p_\theta(\mathbf{x}|\mathbf{z})$, is a good fit to the data (log evidence is large) and the variational distribution is a good approximation to the posterior, $p(\mathbf{z}|\mathbf{x})$.

Having learnt $\theta$ and $\phi$ we can generate images by sampling $\mathbf{z}$ from the latent space according to the prior $p(\mathbf{z})$ and pushing $\mathbf{z}$ through the decoder map, $D_\theta(\mathbf{z}) : \mathbf{z} \to \mathbf{x}$. We can also generate images, $\hat{\mathbf{x}}$, conditioned on test $\mathbf{x}$, in three steps. First by using the encoder map, $E_\phi(\mathbf{x}) : \mathbf{x} \to \mu, \boldsymbol{\Sigma}$, to characterise the posterior, $q_\phi(\mathbf{z}|\mathbf{x}; \mu, \boldsymbol{\Sigma})$. Second, by drawing samples from this distribution. And third, by using the decoder map as before, $D_\theta(\mathbf{z}|\mathbf{x}) : \mathbf{z}|\mathbf{x} \to \hat{\mathbf{x}}$. The full VAE is trained by maximising an evidence lower bound (ELBO) which is equivalent to minimizing the KL divergence between $p(\mathbf{z}|\mathbf{x})$ and $q_\phi(\mathbf{z}|\mathbf{x})$ (Kingma & Welling, 2014), given by

$$
\begin{aligned}
D_{KL}(q_\phi(\mathbf{z}|\mathbf{x}) \parallel p(\mathbf{z}|\mathbf{x})) &= \mathbb{E}_{z \sim q_\phi(\mathbf{z}|\mathbf{x})}[\log q_\phi(\mathbf{z}|\mathbf{x}) - \log p(\mathbf{z}|\mathbf{x})] \\
&\leq \mathbb{E}_{z \sim q_\phi(\mathbf{z}|\mathbf{x})}[\log q_\phi(\mathbf{z}|\mathbf{x}) - \log p_\theta(\mathbf{x}|\mathbf{z}) - \log p(\mathbf{z})] \equiv \mathcal{L}. 
\end{aligned} \tag{1}
$$

### 2.2 Extension of VAE framework to sensor measurements

We now consider the situation where we wish to determine a new $\mathbf{x}$ by taking optimal sequential measurements on $\mathbf{x}$ with respect to a measurement basis with $N$ indexes. At sequential step $k$ the measurement

basis indexes chosen so far are denoted $B^k = \{b_1, b_2, \ldots, b_k\}$. We obtain $\mathbf{y}^{[B^k]}$ by taking a measurement using these basis indices; for convenience we write

$$\mathbf{y}^{[B^k]} = f(\mathbf{x}, B^k). \tag{2}$$

We model these measurements using isotropic Gaussians with variance $\sigma^2$.

### 2.3 SVI method

*Marginalising* $\mathbf{x}$ *and applying Bayes' rule, we have*

$$p(\mathbf{z}|\mathbf{y}) = \frac{p(\mathbf{z})p(\mathbf{y}|\mathbf{z})}{p(\mathbf{y})}. \tag{3}$$

The log of the posterior distribution of the latent variables for a given partial observation $\mathbf{y}^{[B^k]}$ is therefore

$$\log p(\mathbf{z}|\mathbf{y}^{[B^k]}) = \log p(\mathbf{z}) + \log p(\mathbf{y}^{[B^k]}|\mathbf{z}) - \log p(\mathbf{y}^{[B^k]}). \tag{4}$$

This equation, as Equation (2), is intractable motivating the use of SVI. The aim of SVI is to approximate the posterior with a multivariate Gaussian and infer the mean $\mu^k$ and variance $\Sigma^k$ of this distribution. As the posterior, $q^k = q(\mathbf{z}|\mathbf{y}^{[B^k]}; \mu^k, \Sigma^k)$, evolves with the number and index of basis measurements, the mean and variance are indexed by $k$. SVI is an iterative method. The variational parameters for each data input are randomly initialized and then optimized to minimise the KL divergence

$$
\begin{aligned}
D_{KL}(q^k(\mathbf{z}|\mathbf{y}^{[B^k]}) \parallel p(\mathbf{z}|\mathbf{y}^{[B^k]})) &= \mathbb{E}_{\mathbf{z} \sim q^k(\mathbf{z}|\mathbf{y}^{[B^k]})}[\log q^k(z|\mathbf{y}^{[B^k]}) - \log p(\mathbf{z}|\mathbf{y}^{[B^k]})] \\
&\leq \mathbb{E}_{\mathbf{z} \sim q^k(\mathbf{z}|\mathbf{y}^{[B^k]})}[\log q^k(\mathbf{z}|\mathbf{y}^{[B^k]}) - \log p(\mathbf{y}^{[B^k]}|\mathbf{z}) - \log p(\mathbf{z})] \equiv \mathcal{L}^k.
\end{aligned} \tag{5}
$$

### 2.4 Partial encoder

The aim of the partial encoder is to encode incomplete measurements $\mathbf{y}^{[B^k]}$ as defined in Equation (2). We introduce a partial variational distribution, $q_{\phi^p}$, to approximate the posterior distribution and train a partial encoder, $E_{\phi^p}$, to infer $\mu^k$ and $\Sigma^k$ for specific $\mathbf{y}^{[B^k]}$, see Figure 1d). The partial encoder VAE is trained by minimizing the KL divergence between $p(\mathbf{z}|\mathbf{x})$, established by the full VAE, and the partial posterior distribution, $q_{\phi^p}(\mathbf{z}|\mathbf{y}^{[B^k]}; \mu^k, \Sigma^k)$, parameterised also by a neural network with parameters $\phi^p$.

$$
\begin{aligned}
D_{KL}(q_{\phi^p}(\mathbf{z}|\mathbf{y}^{[B^k]}) \parallel p(\mathbf{z}|\mathbf{x})) &= \mathbb{E}_{z \sim q_{\phi^p}(z|\mathbf{y}^{[B^k]})}[\log q_{\phi^p}(\mathbf{z}|\mathbf{y}^{[B^k]}) - \log p(\mathbf{z}|\mathbf{x})] \\
&\leq \mathbb{E}_{z \sim q_{\phi^p}(\mathbf{z}|\mathbf{y}^{[B^k]})}[\log q_{\phi^p}(\mathbf{z}|\mathbf{y}^{[B^k]}) - \log p(\mathbf{x}|\mathbf{z}) - \log p(\mathbf{z})] \equiv \mathcal{L}_{partial}.
\end{aligned} \tag{6}
$$

As with the full VAE, once we have learnt $\phi^p$ we can generate images from a set of measurements by sampling from the approximate posterior distribution $q_{\phi^p}(\mathbf{z}|\mathbf{y}^{[b]})$ and then pushing these samples through the decoder, $D_\theta(\mathbf{z}|\mathbf{y}^{[B^k]}) : \mathbf{z}|\mathbf{y}^{[B^k]} \to \hat{\mathbf{x}}$ to generate the reconstructed image $\hat{\mathbf{x}}$.

In summary, we have two approaches to approximate the posterior $p(\mathbf{z}|\mathbf{y}^{[B^k]})$. First using a partial encoder, $q_{\phi^p}(\mathbf{z}|\mathbf{y}^{[B^k]})$, and second using SVI, $q^k(\mathbf{z}|\mathbf{y}^{[B^k]})$. We compare these approaches in Section 4.2.

### 2.5 Training the partial encoder

To train the partial encoder we assume that our measurement sensor can provide a series of $J = B^k$ observations, $\{y_1, y_2, \ldots, y_J\}$, each associated with an action (experiment or basis measurement resulting in

an observation) for an image $\mathbf{x}$, see Equation (2). We now simulate training data by randomly sampling $N$ experiments for every image $\mathbf{x}$, simulating observations $\mathbf{y}^{[J]}$ for each of them. This is repeated $E$ times for each $\mathbf{x}$ giving $N \times E$ experiments in total where the measurement vector, $\mathbf{y}^{[J]}$, varies in terms of both the number of measurements and the measurement index. This is achieved in an efficient manner by introducing a mask layer into the encoder network that randomly masks a different number and index of measurements with each training batch. Using this training set we learn a variational autoencoder which generates the required mapping, $E_{\phi^p}(\mathbf{y}^{[J]}, j) :\to \mu^J, \Sigma^J$.

## 3 Method for active sequential algorithm

### 3.1 The Active Sequential Measurement Inference Algorithm

We now present our sequential algorithm to actively choose the next best measurement. Our proposed active inference algorithm is illustrated in Figure 1 and pseudocode provided in Algorithm 1. The aim of the algorithm is to identify, under some criteria (details in Section 3.2), the next best measurement to take. The algorithm is designed to leverage the encoding and decoding properties of the VAE. The encoder provides a means to map incomplete measurements onto a low dimensional space for exploration. The decoder provides a means to project back to image space to assess future measurements.

At the first step, $k = 0$, $N$ samples are drawn from the prior on the latent space, $p_0 = \mathcal{N}(0, 1)$. These samples, $\mathbf{z}_i$, are pushed through the decoder, $D_\theta(\mathbf{z}_i) : \mathbf{z}_i \to \hat{\mathbf{x}}_i$, to obtain $N$ generated images, $\hat{\mathbf{x}}_i$. Simulated measurements, $\hat{\mathbf{y}}_i = f(\hat{\mathbf{x}}_i, j)$, are made under the chosen measurement basis for each basis element indexed $j$. The simulated measurements are pushed through the encoder, $E_{\phi^p} : \hat{\mathbf{y}}_i \to: \mu^j, \Sigma^j$, and the output used to characterise the conditional posterior, $q_i^j$. The algorithm evaluates each measurement indicator and chooses the measurement indicator which best satisfies the decision criteria. This indicator is added to the indicator set, $B^{k+1}$. Actual measurements are taken on the test image, $\mathbf{y}^{[B^{k+1}]} = f(\mathbf{x}, B^{k+1})$. SVI is used to infer the posterior and predictive prior for the next step, $p^{k+1} = q^{k+1}$. The algorithm continues for $K - 1$ steps.

At step $k + 1$, $N$ samples, $\mathbf{z} = \{z_1, \ldots z_N\}$, are drawn from the predictive prior, $p_k = \mathcal{N}(\mu^{B^k}, \Sigma^{B^k})$, where $\mu^{B^k}$ and $\Sigma^{B^k}$ are the mean and variance predicted by the partial encoder conditional on the measurements, $\mathbf{y}^{[B^k]}$, made so far

$$\text{enc}_{\phi^p}(\mathbf{y}^{[B^k]}) : \mathbf{y}^{[B^k]} \to \mu^{B^k}, \Sigma^{B^k}. \tag{7}$$

These samples are then mapped by the decoder to generate images

$$\text{dec}_\theta(\mathbf{z}) : \mathbf{z} \to \hat{\mathbf{x}} = \{\hat{x}_1 \ldots \hat{x}_N\}. \tag{8}$$

For each remaining pattern indicator, $j \in B \backslash B^k$, we create a set of pattern indicators $J = \{b_1, \ldots b_k, b_j\}$ and simulate measurements made on each generated image, $\{\hat{\mathbf{y}}_1^{[J]}, \ldots, \hat{\mathbf{y}}_N^{[J]}\}$. We now use the partial encoder to approximate the posterior distribution conditional on these simulated measurements for each generated image denoted $q_i^J = \mathcal{N}(\mu_i^J, \Sigma_i^J)$.

### 3.2 Criteria for choosing next measurement

We consider three criteria for choosing the next pattern to measure. A good choice of measurement is one that provides useful information to the agent.

#### 3.2.1 Likelihood (QP)

Here, the criterion for choosing the next pattern, $b_{k+1}$, corresponds to choosing the pattern, $b_j$, with the highest log likelihood with respect to $q_i^J$ over all $N$ images;

$$b_{k+1} = \arg\max_j \sum_{i=1}^{N} \log(q_i^J(z_i)) - \log(p_k(z_i)). \tag{9}$$

By using the likelihood we establish which measurements provide information that is consistent with the predictive prior for that step. We investigate the ability of the algorithm to move towards the target distribution.

### 3.2.2 Mutual Information (MI)

An alternative criterion based on the concept of conditional mutual information Bishop (2006) is to choose the measurement which most reduces the posterior entropy or uncertainty. The mutual information, $MI$, between $\mathbf{z}_i$ and $\hat{\mathbf{y}}_i^{[J]}$ is defined in terms of entropy $H^e$ as

$$MI(\mathbf{z}_i; \hat{\mathbf{y}}_i^{[J]}) = H^e(\mathbf{z}_i) - H^e(\mathbf{z}|\hat{\mathbf{y}}_i^{[J]}). \tag{10}$$

The entropy of random variable $\mathbf{x}$ from a multivariate Gaussian of dimension $D$ and variance $\Sigma$ is

$$H^e(\mathbf{x}) = \frac{D}{2}\left(1 + \log\left(2\pi\right)\right) + \frac{1}{2}\log|\Sigma|. \tag{11}$$

Our criteria for choosing the next pattern is then

$$b_{k+1} = \arg\max_j \sum_{i=1}^{N} \frac{1}{2}(\log|\Sigma_i^J| - \log|\Sigma^{B^k}|). \tag{12}$$

### 3.2.3 Inference-free Hadamard optimisation (HO)

Some measurement bases, for example the Hadamard transform (Ahmed & Rao, 1975), permit patterns to be prioritised according to the absolute value of the measurement instead of involving inference. High absolute values (of the eigenvalues associated with each basis eigenvector) contribute more to the reconstructed image which motivates the choice

$$b_{k+1} = \arg\max_j \sum_{i=1}^{N} |\{\hat{\mathbf{y}}_i^{[J]}\}|. \tag{13}$$

## 4 Experimental Results

We illustrate the algorithm on Fashion MNIST (Xiao et al., 2017). The basic VAE is trained using the 60,000 training images from Fashion MNIST Xiao et al. (2017). The encoder/decoder architectures and training details are provided in Appendix A. The partial encoder is trained on simulated measurements from a novel 4×4 convolutional Hadamard measurement basis, details given in Section 4.1. The partial encoder architecture is adapted and fitted with a random measurement layer so that experiments involving different numbers and types of patterns can be efficiently simulated, in terms of computation and memory, during training.

### 4.1 Convolutional Hadamard basis

In the context of single pixel imaging, the measurements, $\mathbf{y}$, are made by projecting a series of spatial patterns on to a scene and capturing the reflected light with a single pixel detector sensor Higham et al. (2018). Mathematically, the measurement is the inner product between the patterns and the scene and defines function $f$ from equation (2). Expressing an image in vector form, $\mathbf{x}$, and the pattern basis in matrix form, $\mathbf{H}$, where $\mathbf{H} \in \mathbb{R}^{D \times D}$, we have $\mathbf{y} = \mathbf{H}\mathbf{x}$. Of particular interest is the Hadamard basis, an orthogonal binary $-1, 1$ basis (Ahmed & Rao, 1975). A Hadamard basis is suitable for experimental realization due to the binary nature of patterns that can be projected using DMD (Digital Micromirror Device) technology as spatial light modulator (Edgar et al., 2019). We take $\mathbf{H}$ to be this basis though we emphasise our method is not specific to the Hadamard basis and the approach could be generalised to another appropriate basis. For an image with $2^n \times 2^n$ pixels where $n$ is a positive integer, the complete Hadamard basis, required for perfect image reconstruction, comprises $N = 2^{2n}$ patterns.

---

**Algorithm 1** Active Sequential Inference

---

$B^0 \leftarrow \emptyset$          ▷ pattern index set
**for** $k = 0 \ldots K - 1$ **do**          ▷ for each step
    **for** $i = 1 \ldots N$ **do**          ▷ for each generated image
        **if** $k = 0$ **then**
            $p_0 = \mathcal{N}(0, 1)$          ▷ set prior for step 0
        **else if** $k > 0$ **then**
            $p_k \leftarrow q^k$          ▷ set predictive posterior prior for step $k$
        **end if**
        $\mathbf{z}_i \sim p_k$          ▷ sample latent vector from current pdf
        $\hat{\mathbf{x}}_i \leftarrow \text{dec}_\theta(\mathbf{z}_i)$          ▷ **A.** input to decoder to obtain generated image $\hat{\mathbf{x}}_i$
        **for** each element $j \in B \backslash B^k$ **do**          ▷ for each remaining pattern
            $J \leftarrow \{b_1, \ldots, b_k, j\}$          ▷ add pattern index $j$ to measurement set $J$
            $\hat{\mathbf{y}}_i \leftarrow f(\hat{\mathbf{x}}_i, J)$          ▷ **B.** estimate possible measurements
            $q_i^j \leftarrow \text{enc}_{\phi^p}(\hat{\mathbf{y}}_i)$          ▷ **C.** use partial encoder to approximate pdf $q^j$
        **end for**
    **end for**
    $\mathbf{M}_{ij} \leftarrow \log(q_i^j(\mathbf{z}_i)) - \log(p_k(\mathbf{z}_i))$          ▷ **D.** store results in matrix $\mathbf{M}$
    $\{b_{k+1}\} \leftarrow \{\arg\max_j \sum_i \mathbf{M}_{ij}\}$          ▷ **E.** find pattern index which maximises expression
    $\mathcal{B}^{k+1} \leftarrow \mathcal{B}^k \cup \{b_{k+1}\}$          ▷ add pattern index to pattern index set
    $\mathbf{y}^{[B^{k+1}]} \leftarrow f(\mathbf{x}, B^{k+1})$          ▷ take actual measurement on $\mathbf{x}$
    $q^{k+1} \leftarrow SVI(\mathbf{y}^{[B^{k+1}]})$          ▷ **F.** set predictive posterior prior for next step
**end for**

---

In this work we develop a novel convolutional Hadamard basis inspired by convolutional layers. The convolutional approach provides local spatial and resolution rather than global frequency information.

A Hadamard basis matrix with $2^4$ rows and columns is rearranged as a $4 \times 4 \times 16$ tensor which replaces the filter of a standard convolutional mapping layer, $f_{conv}$. The Hadamard basis has the property of being its own inverse so the tensor can be used with transpose convolutional mapping layer, $f_{tpconv}$, to recover the input image from the feature image, $f_{conv} : \mathbf{x} \to \mathbf{y}$ and $f_{tconv} : \mathbf{y} \to \mathbf{x}$.

An advantage of this convolutional Hadamard basis is that the resulting feature $f \in \mathbb{R}^{N/4 \times N/4 \times 16}$ has both spatial (vertical and horizontal) and frequency or resolution dimension associated to each element $f_j$ where $j = 1 \ldots N^2$. We exploit this in our illustration to give further insights into the decision making process.

### 4.2 Comparison of pVAE and SVI

At each step of the algorithm, estimating the parameters of the approximate posterior, $q^J$, requires one pass through the partial encoder but several iterations with the SVI method. We compare the performance of pVAE (1 iteration) with SVI and a variable number of iterations $\{10, 20, 30, 40, 50, 60, 70\}$ in terms of reconstruction indexes: mean square error (MSE) and similarity structure (SSIM) Wang et al. (2004). The performance scores are averaged over 100 samples from ten images (each from a different class) and the algorithm is run for 100 steps, see Figure 3. The SVI method improves as the number of iterations increases but only matches the pVAE method after 60 iterations. This makes the SVI method an order of magnitude slower than pVAE. For our final algorithm, we therefore use pVAE to approximate $q_i^J$ based on simulated measurements (C in Algorithm 1) and SVI with 100 iterations to approximate $q^{k+1}$ based on actual measurements (F in Algorithm 1). This way we achieve a balance between performance and computation time.

### 4.3 Comparison of choice criteria

The different criteria for choosing the next pattern (**QP**, **MI** and **HO**), equations (9), (12) and (13) respectively, were evaluated using 10 test images, one from each class, and $Ns = \{1, 10, 100, 200\}$ latent vector

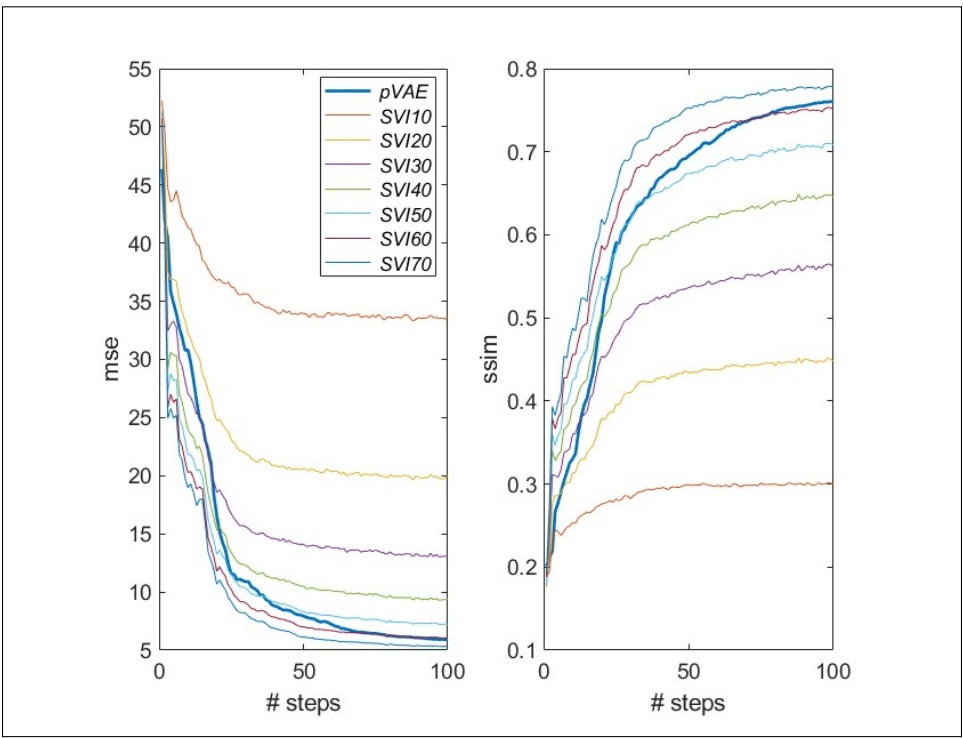

Figure 3: Comparison of pVAE and SVI. The performance scores (mse *left column* and ssim *right column*) are averaged over 100 samples from ten images (each from a different class) and the algorithm is run for 100 steps. Here lower is better for mse *left* and higher is better for ssim *right*. The performance of pVAE (1 iteration) *bold line* with SVI and a variable number of iterations $\{10, 20, 30, 40, 50, 60, 70\}$ *mixed light lines. The SVI method improves as the number of iterations increases. At 100 steps the results for pVAE lie between the results for SVI with 60 (SVI60) and 70 (SVI70) iterations. In terms of timings, the pVAE takes $9 \times 10^{-4}$ seconds per step and the SVI method takes $7.2 \times 10^{-3}$ seconds per iteration. With many iterations required for SVI, this makes the SVI method at least an order of magnitude slower than pVAE. Time measurements were taken using a NVIDIA GE Force RTX 3090 GPU.*

samples over 100 steps. Performance measures, SSIM and MSE, were evaluated at each step and averaged over the test images, see Figure 4 and, in Appendix B, Figure 6.

Our method (**QP**) outperforms Hadamard optimisation (**HO**) for the first 25 steps with 1, 10, 100 and 200 starting images in terms of MSE and SSIM. The methods **QP** and **HO** differ in both the criteria for choosing the next pattern and reconstruction whereas the methods **QP** and **MI** differ only in the criteria for choosing the next pattern. After these first steps, the performance of both methods plateaus with a slightly superior performance from the Hadamard reconstruction. Comparing **QP** with **MI**, **QP** shows superior performance across all the experiments. Investigation of the patterns chosen suggest that using **MI** early in the process leads to the algorithm getting stuck in the latent space. This method relies on choosing the next pattern to reduce uncertainty across several generated images whereas **QP** chooses the next pattern to increase likelihood. The **MI** strategy promotes larger initial steps in the latent space and consequently a tendency to get stuck and miss patterns that are useful for reconstruction. The **QP** strategy encourages smaller steps in the latent space and moves more steadily to a convergence of generated images. The **QP** method is therefore the one that we recommend.

### 4.4 Visualisation of the active learning algorithm using UMAP

We now illustrate the active learning algorithm by exploring low dimensional image space using a two dimensional representation of the latent space of 10,000 test images from Fashion MNIST created with

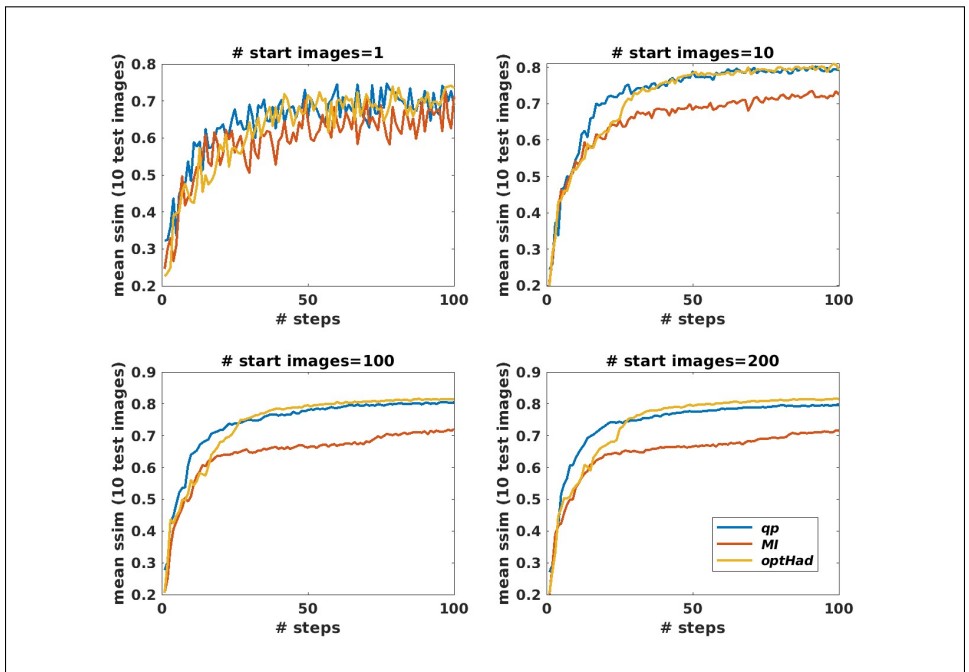

Figure 4: Comparison of choice criteria in terms of the structural similarity index (SSIM). The different criteria for choosing the next pattern (**QP**, **MI** and **HO**), equations (9), (12) and (13) respectively, were evaluated (mean SSIM) using 10 test images, one from each class, and 1 (*top left*), 10 (*top right*), 100 (*bottom left*), 200 (*bottom right*) latent vector samples over 100 steps. Our method (**QP**) outperforms Hadamard optimisation (**HO**) for the first 25 steps and both show superior performance to **MI** across all the experiments. *In terms of timings, **HO** took 0.7778 seconds, **QP** took 1.3583 seconds and **MI** took 1.4219 seconds to complete 50 steps. Time measurements were taken using a NVIDIA GE Force RTX 3090 GPU.*

UMAP Meehan et al. (2022). UMAP is applied to the mean of the latent representations. The classes are colour coded and the UMAP clusters within class and between similar classes (i.e. ankle boot, sneaker and shoe) indicate that class structure has been retained by the latent representation and further dimension reduction, see Figure 5.

A number, $N$, of latent variables are drawn from the prior. These are shown as black crosses projected on to the two dimensional space along with the projected class colour coded test images. These variables are then passed to the generator, $p_\theta(\mathbf{x}_i|\mathbf{z}_i)$, to produce the generated images shown on the right. Also shown is the image reconstruction from actual measurements made so far (*top right box*), none at step 0, 10 at step 10 and 30 at step 30, and the target image (*bottom left box*).

At steps 1 to $K$, the generated images from the previous step are used to estimate possible measurements corresponding to the set of pattern indexes not yet taken. These possible measurements are individually added to the actual measurements and passed to the encoder to obtain the posterior probability distribution. The measurement index, $j$, which is considered most informative satisfies the following expression:

$$\arg\max_j \sum_i \log(q_{ij}(z_i) - \log(p_k(z_i)). \tag{14}$$

At step $k$ our certainty, quantified by a probability distribution about our location in latent space, having taken $k-1$ measurements, is denoted by $p_{k-1}(z_i)$. If we were to take another measurement, our certainty becomes $q_{kij}(z_i)$. In the interest of increasing our certainty we choose the value which maximises the above expression. This simple procedure allows us to move through latent space converging on a reconstruction close to the target.

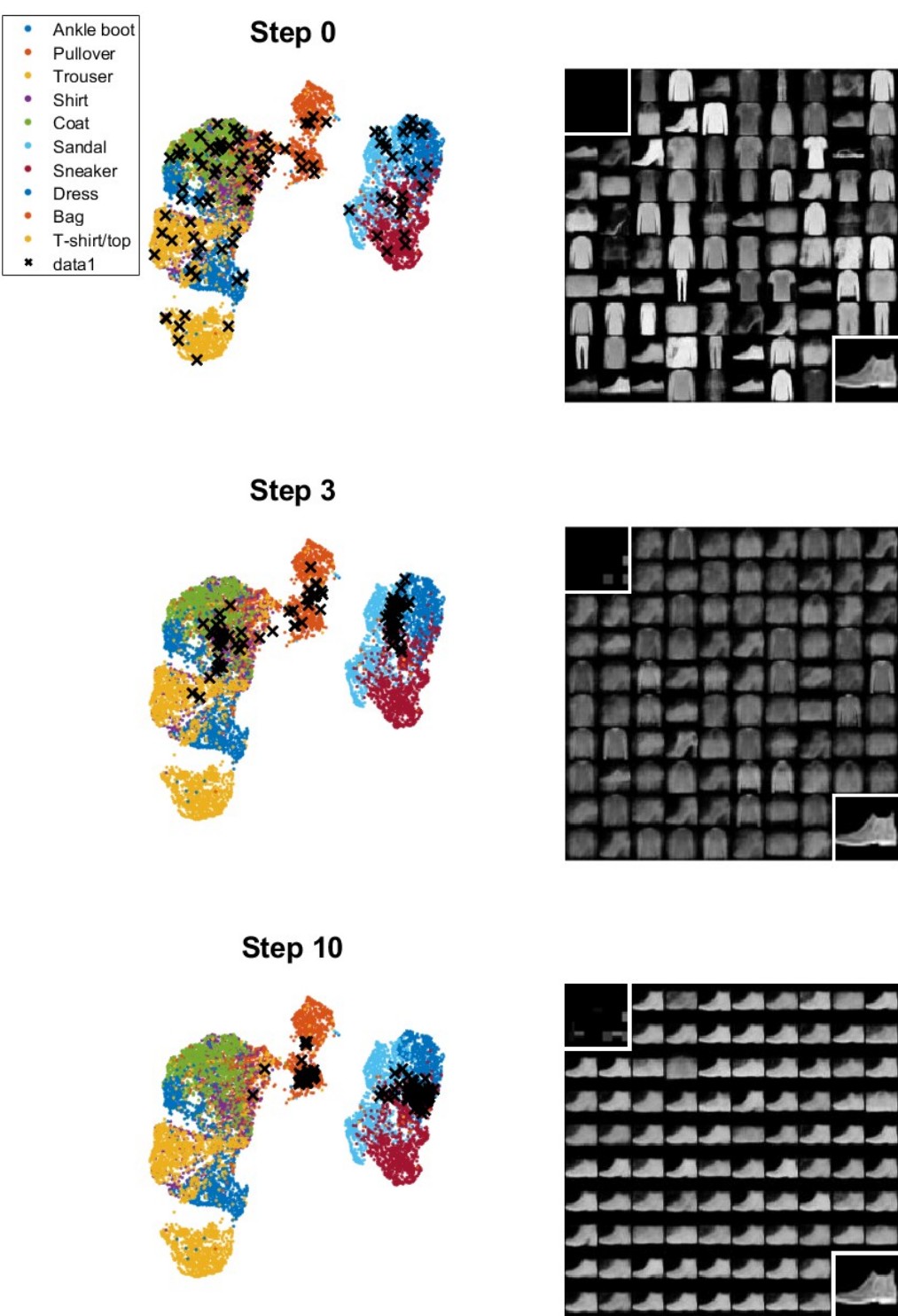

Figure 5: UMAP is used to reduce the mean of the latent representations of 10,000 test images to 2 dimensions (*left hand column*). The classes are colour coded and the UMAP clusters within class and between similar classes (i.e. ankle boot, sneaker and shoe) indicate that the class structure has been retained by the latent representation and further dimension reduction. The mean of 100 samples is similarly projected (black crosses) on to the map at each step of the algorithm. The measurements made, the samples projected back into image space and the image to be recovered are shown in the top left corner, centre and bottom right corner of the (*right hand column*) respectively. We see the diversity of possible images in the *right hand column*, and this decreases as uncertainty is reduced by more measurements.

## 5 Discussion and Conclusion

We have shown that given a large dataset and a measurement basis the encoder of a VAE, trained on this large dataset, can be adapted to map partial measurements on to a representative latent space. An sequential measurement algorithm is developed to explore this latent space in order to optimise the next best measurement. The algorithm is illustrated using the Fashion MNIST dataset and a novel convolutional Hadamard measurement basis. We see that useful patterns are chosen within 10 steps leading to the convergence of the guiding generative images. In situations where there is a cost attached to measurement, the ability to reduce the number of measurements is a significant benefit. We believe that this algorithm is the first to address the task of active sequential inference in this context, and we note that it has the potential to increase efficiency dramatically in many high profile applications.

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

## A  VAE architecture and training

The encoder comprised an image input layer, two encoding blocks and a fully connected layer. Each encoding block contained a convolutional layer, a batchnorm layer and a ReLU activation layer. The input image $(28 \times 28)$ was downsized to $(14 \times 14 \times 32)$ and $(7 \times 7 \times 64)$ by the encoding blocks respectively. The output of the fully connected layer, $[\mu, \log \Sigma]$, is twice the size of the latent space $(32 \times 1)$.

The partial encoder was formed by replacing the first encoding block with a convolutional layer, modified to use the convolutional Hadamard basis as fixed weights, resulting in a feature $(7 \times 7 \times 64)$. This block was followed by a random mask layer that randomly selects a number of patterns and pattern indexes. The subsequent encoding block was adjusted to downsize to $(4 \times 4 \times 64)$.

The decoder comprised an latent sample input layer $(16 \times 1)$, a project and reshape layer $(7 \times 7 \times 64)$ and three decoding blocks. The decoding blocks each contained a transposed convolutional layer and an activation layer RELU for the first two blocks and sigmoid for the last block). The input feature was up sampled to $(14 \times 14 \times 64)$, $(28 \times 28 \times 32)$ and $(28 \times 28 \times 1)$ by the decoding blocks respectively.

The encoder and decoder were trained together using a custom training loop and 60,000 fashion MNIST images Xiao et al. (2017) in mini-batches of 128 for 100 epochs. The parameters were updated using the adaptive moment estimation (ADAM) algorithm *(Kingma & Ba, 2015)* with settings: learning rate = 0.001, gradient decay = 0.9, squared gradient decay = 0.999 and epsilon = 1e-8) chosen using validation set performance.

The partial encoder was trained using the previously trained decoder with fixed settings for 200 epochs. The random mask layer was reset for each mini-batch iteration simulating $200 \times 450$ different experiments.

We modify the KL divergence term in the loss functions to include an additional scaling factor, $\beta$, in front of the KL divergence term. This $\beta VAE$ approach was introduced in Higgins et al. (2017) to encourage a more flexible latent space representation, while still ensuring that the learned distribution is close to the prior distribution. The value of $\beta$ is set to 0.1 for the VAE and the partial VAE.

## B  More results

The different criteria for choosing the next pattern (**QP**, **MI** and **HO**), equations (9), (12) and (13) respectively, were evaluated using 10 test images, one from each class, and $Ns = \{1, 10, 100, 200\}$ latent vector samples over 100 steps. Performance measures, SSIM and MSE, were evaluated at each step and averaged over the test images, see Figure 4 and, in Appendix B, Figure 6.

## C  Interpreting results

The patterns belonging to the convolutional Hadamard basis have two spatial and one resolution component. A log likelihood map for each generated image can be formed by averaging the row $M_{i,:}$ over the resolution component

$$M_{i,xy} = \sum_r M_{i,xyr}. \tag{15}$$

Figure 7 shows resized $M_{xy}$ overlaid on the generated image $\hat{x}$ at step 0. This visualisation highlights regions of interest. Namely the tops of the sleeves for T-shirt (a), the back and toe of the boot (b) and the waist and lower legs for the trousers (c).

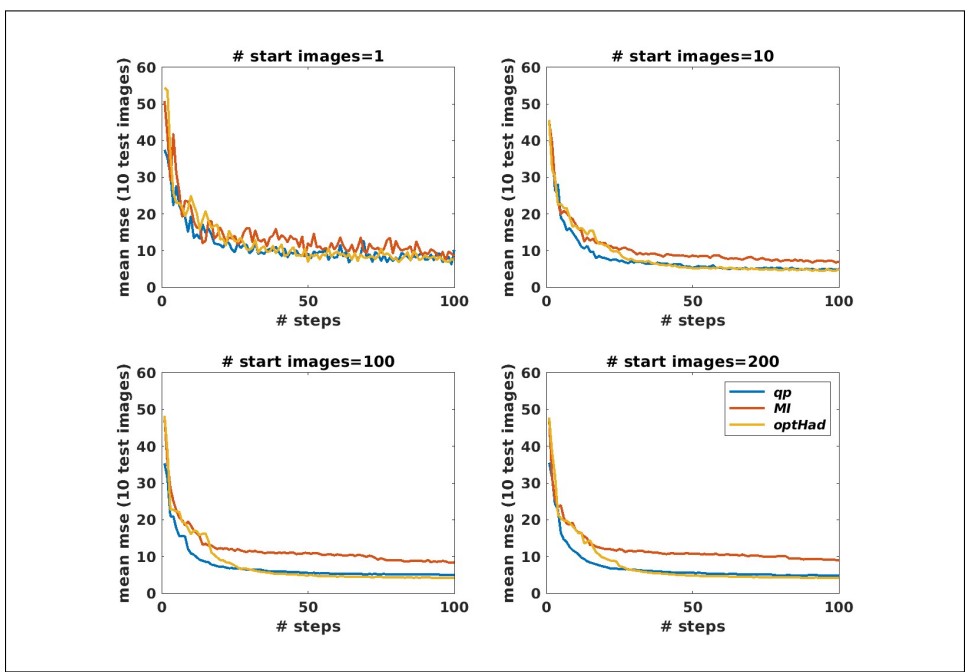

Figure 6: Comparison of choice criteria in terms of mean squared error (MSE). The different criteria for choosing the next pattern (**QP**, **MI** and **HO**), equations (9), (12) and (13) respectively, were evaluated (mean MSE) using 10 test images, one from each class, and 1 (*top left*), 10 (*top right*), 100 (*bottom left*), 200 (*bottom right*) latent vector samples over 100 steps. (**QP**) outperforms Hadamard optimisation (**HO**) for the first 25 steps.

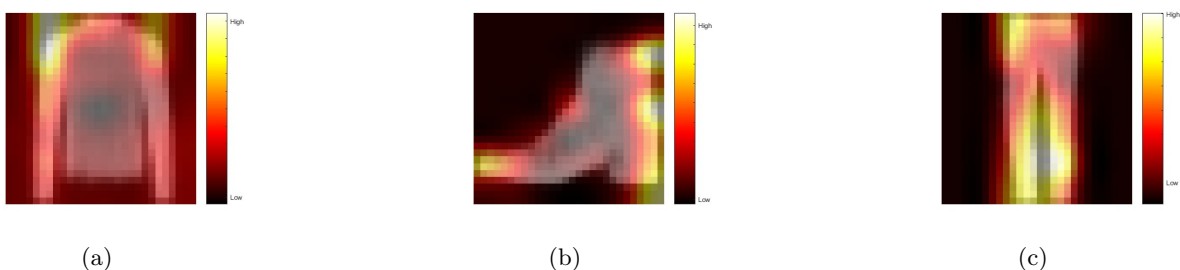

Figure 7: After step 0, we overlay $M_i$ over $\hat{x}_i$ to indicate regions of high information (white) and low information (black) within the active learning frame at this step. For the long sleeved top, regions of interest are the tops of the sleeves (a). The back and toe of the boot are regions of interest (b). The waist and lower legs are regions of interest for the trousers (c). Using expression in equation 14 the next measurement taken is determined by averaging the information over $N$ images.

