# OpenReview forum: "Actively Inferring Optimal Measurement Sequences"
_TMLR — Rejected by TMLR_

### Review · Reviewer_LXw8 · 2024-09-21

**Summary Of Contributions:**

The paper proposes using a VAE with a partial encoder for an active sequential inference algorithm.
It works by learning the clean representation of a partial observation and uses that to select the next measurement based on certain criteria.
The work compares two inference of the clean representation: one based on SVI and one based on amortized inference.
3 types of index selection criteria, likelihood (difference), mutual information and inference-free Hadamard optimization, are discussed.
Empirical study on FashionMNIST is provided to demonstrate the method quantitively and UMAP is used to show how it works qualtitively.

**Audience:**

Yes

**Claims And Evidence:**

No

**Requested Changes:**

Some writing issues:
- There are missing integrals in (3).
- Inconsistent bold v.s. normal $y$ in (5) and (6).
- Extra $p$ in the KL part of (6).
- Title of section 3: "Method" -> "Method for" perhaps?
- (7) and (8) please either use `\text` or `\mathrm` for `enc` and `dec`.

Please also refer to [Requested] in the Weakness and Questions of the previous section.

**Strengths And Weaknesses:**

## Strengths
- The paper is well-written and easy to follow.
- The method is sound, novel and simple but effective.

## Weaknesses
- The empirical study is over-simplified.
  - [Requested] At least some dataset that is really used in such active measurement case should be used to validate the proposed method.

## Questions
- Why can't you use (5) to learn the partial encoder? Seems that it's a missing alternative.
- A practical concern of SVI vs amortized inference is that do we have that many clean data in those measurement problems to even learn the encoder? [Requested] I think some study on how the amount of available data affect the overall method as well as the choice between SVI and encoder is needed, given the method is motivated to solve the problem when measurement is expansive.
- How is the conditional entropy in (10) estimated?

---

> ### Author Response · Authors · 2024-10-11
> **Authors' response**
>
> We are pleased that you found the paper well-written and easy to follow. Thanks for pointing out the minor issues in writing; we will correct these in the revised version.
> You mention that the method is sound, novel and simple but effective. However over-simplified and gave a negative response to evidence.
> Please can we clarify which of our claims are unsupported by the evidence we present?  Our objective was to present the core method as clearly as possible as the basis for future experimental publications. Introducing further relevant real-world scenarios would necessitate significant space for further explanation and detract from our key message, explaining the core algorithm. We would argue that we have provided evidence that the method works for example dataset chosen to illustrate it and will be able to share code for anyone to confirm this. We are happy to add a sentence as a caveat that further future tests on a wider range of problems will enhance our knowledge of its strengths and limitations.
>
> Concerning your questions:
> The loss function for the SVI method, equation (5) uses the measurements as regularisation whereas the loss function for the partial encoder uses the output of the decoder as regularisation. Using (5) instead of (6) would decouple the encoder from the decoder. This moves away from our objective, of including information from the full VAE latent representation, so we do not consider it as an alternative method.
> Please can you clarify ‘available data’? Do you mean for training or testing? The expressivity of the full VAE will depend on training data. The inference starts from a prior which is updated at each step based on available measurements. The partial encoder, unlike SVI, has been exposed to a range of available measurements. For relevant real-world domains where compressed sensing takes place, clean observations are typically easily available for training. The comparison in terms of performance will depend on how well the partial encoder can map to the latent space.
> The conditional entropy in equation (10) is defined in terms of the dimension of the latent space (D) and the variance $\Sigma_i^J$ as in equation (11). We will add brackets to equation (12) to make this clearer.

---

### Review · Reviewer_FJyZ · 2024-10-01

**Summary Of Contributions:**

This paper uses variational methods for active sequential inference. It focuses on recovering high dimensional data by making as few measurements as possible. They compare stochastic variational inference (SVI) and a partial variational auto-encoder (pVAE) for the task of image reconstruction using the Fashion MNIST dataset. They compare three criteria for choosing the next measurement (likelihood, mutual information, and inference-free Hadamard optimization) and find that inference-free Hadamard optimization performs best across several numbers of starting images.

**Audience:**

Yes

**Claims And Evidence:**

Yes

**Requested Changes:**

* Consider adding plots that show the runtime of each method.
* Consider clarifying the difference between variational inference and inference (e.g., “inference-free”).
Requested Clarifications:
* In Equation (6), should $D_{KL}(qp_{\phi^p}(z | y^{[B^k]} || p(z | x))$ be $D_{KL}(q_{\phi^p}(z | y^{[B^k]} || p(z | x))$?

**Strengths And Weaknesses:**

Strengths:
* The paper gives a nice introduction to variational auto-encoders (VAEs), including graphical representations (see Figure 2).
* The paper includes a nice comparison of criteria for choosing the next measurement across several numbers of starting images.

Weaknesses:
* Experiments are on only one dataset.
* "The SVI method improves as the number of iterations increases but only matches the pVAE method after 60 iterations." Is this true? In Figure 3, the SVI with 70 iterations has lower MSE and higher SSIM than the pVAE.
* "This makes the SVI method an order of magnitude slower than pVAE." The computational difference between SVI and pVAE is unclear. Does pVAE use the same computation as SVI with 1 iteration? Or does the pVAE model add computational complexity?

---

> ### Author Response · Authors · 2024-10-11
> **Authors' response**
>
> We are pleased that you found the introduction to VAEs and criteria outlined in the paper strong. It is our aim to present the method as clearly as we can. Our focus has been on using an easily visualised and well-known data for didactic reasons to explain the approach. We wanted to show the effectiveness of our method. We chose Fashion MNIST to illustrate our method, represent performance on an image dataset and show that our method works well on these types of images. Adding a further dataset with a serious closed-loop context would necessitate significant space for further explanation and reduce the number of pages we can use to explain our method.
> We also note that related work in this area such as [7] has used simpler, and less relevant tabular data. Hence, we regard our computations as an advancement on the state-of-the-art.
>
> [7] Lewis, Sarah, et al. "Accurate imputation and efficient data acquisition with transformer-based vaes." NeurIPS 2021 Workshop on Deep Generative Models and Downstream Applications. 2021.
>
> Regarding your questions:
> From the perspective of 100 steps (end of the x-axis) of the algorithm the pVAE method curve lies between the SVI60 (60 iterations at each step) curve and the SVI70 (70 iterations at each step) curve so this led to the original comment. We see that this does not fully consider some activity at 50 steps (MSE left column) so we will clarify our statement. Our overall message is that the pVAE method plateaus between SV160 and SVI70 so overall requires far fewer iterations.
> The comparison is made in terms of iterations, and we will further support our claim by showing the timings. The pVAE does use the same amount of computation as SVI and the complexity with 1 iteration is very similar. We will add a plot to show the runtime of each method.
>
> We will clarify the difference between variational inference and inference in the introduction section when first introducing these terms.
> Equation (6) contains a typo which we will correct.

---

### Review · Reviewer_JGk7 · 2024-10-08

**Summary Of Contributions:**

The paper introduces a novel active sequential inference algorithm designed to minimize the number of measurements needed to recover high-dimensional data efficiently. The VAE maps partial measurements to a low-dimensional latent space conditioning on
the partial measurements, enabling the selection of the most informative next measurement. The algorithm samples from this latent space and uses the VAE decoder to generate data conditioning on these partial measurements. It then evaluates potential measurements before acquisition by estimating them on generated data. The last step is set predictive posterior prior for the next acquisition step. Demonstrated using the Fashion MNIST dataset with a novel convolutional Hadamard measurement basis, the algorithm selects useful measurement patterns within 10 steps, leading to rapid convergence of generative images. Compared to stochastic variational inference to infer posterior for each data individually, this approach processes batches of generated data more efficiently and achieves superior results with minimal measurements. This work is the first to address active sequential inference in the context of VAE conditioning on partial measurements, offering significant benefits in scenarios where measurement costs are high.

**Audience:**

Yes

**Broader Impact Concerns:**

The authors should add discussions on whether the VAE will encode pre-existing societal bias that could further influence which measurements to make, and how such bias may further be exacerbated by the proposed actively inferring optimal measurement approach.

**Claims And Evidence:**

Yes

**Requested Changes:**

Related work could be strengthened. There are many related topics, including active feature acquisition [1], active multimodal acquisition [2], active perception [3], bayesian active learning [4], active-RL [5], active vision [6], and more. [7] is very relevant and the reviewer encourages the authors to discuss it. The reviewer only listed one paper for each topic, but encourages the authors to discuss more.

[1] Saar-Tsechansky, Maytal, Prem Melville, and Foster Provost. "Active feature-value acquisition." Management Science 55.4 (2009): 664-684.

[2] Kossen, Jannik, et al. "Active acquisition for multimodal temporal data: A challenging decision-making task." arXiv preprint arXiv:2211.05039 (2022).

[3] Bajcsy, Ruzena, Yiannis Aloimonos, and John K. Tsotsos. "Revisiting active perception." Autonomous Robots 42 (2018): 177-196.

[4] MacKay, David JC. "Information-based objective functions for active data selection." Neural computation 4.4 (1992): 590-604.

[5] Andreopoulos, Alexander, and John K. Tsotsos. "A theory of active object localization." 2009 IEEE 12th International Conference on Computer Vision. IEEE, 2009.

[6] Whitehead, Steven D., and Dana H. Ballard. "Active perception and reinforcement learning." Machine Learning Proceedings 1990. Morgan Kaufmann, 1990. 179-188.

[7] Lewis, Sarah, et al. "Accurate imputation and efficient data acquisitionwith transformer-based vaes." NeurIPS 2021 Workshop on Deep Generative Models and Downstream Applications. 2021.

**Strengths And Weaknesses:**

Strengths:

- Soundness: This paper is technically sound with clear formulations of the measurement process, VAE for measurement, SVI, partial encoder, the measurement inference process, and the criteria. Derivations in Section 2 seem correct. Experiments using Fashion MNIST is nicely designed with a novel Hadamard basis for simulated measurements. The framework effectively addresses the introduced problem.

- Clarity: This paper is well-written, providing practical motivation, a precise algorithm outline, and helpful visualizations.

- Relevance: This submission is relevant to scenarios where data acquisition is expensive; using a generative model to compare measurements in latent space from partial prior measurements is both useful and interesting.

Weaknesses:

- The related work section omits important prior studies (see requested changes).

- It is unclear if the method generalizes well to larger-scale settings like images in the wild, videos, or long-form texts.

---

> ### Author Response · Authors · 2024-10-11
> **Authors' response**
>
> We are pleased that you find the paper sound, clear and relevant. The task of optimally acquiring high dimensional data is challenging. We show our method working in an image context. The current submission focusses on the algorithmic aspects of our methods with a proof of concept on realistic data which is well understood by the community, and easily visualised for didactic purposes. We are engaged in ongoing collaboration on novel physical experiments with optics researchers (and the challenges of these experiments were the motivation for the submitted research). Setting up and describing these experiments would take us beyond the journal’s page limit and would involve further collaborators. Novel measurement is a key part of our method and leveraging this will enable us to tackle larger-scale settings.
>
> We will be happy to add the suggested related works. The workshop paper [7] is relevant and extends work by Ma et al. 2019, referenced in the paper, by extending the partial VAE architecture with transformer components. One advantage of our partial encoder, over these works, is that given a full VAE on a domain, the partial encoder can be trained on different measurement basis.

---

> > ### Comment · Reviewer_JGk7 · 2024-10-16
> > **Response**
> >
> > The reviewer thanks the authors for the response. It mostly addresses the concerns from the reviewer, and the reviewer maintains the positive rating.

---

### Author Response · Authors · 2024-11-04
**Revised version**

With the next deadline approaching and having received no further feedback from two of the reviewers, we now submit a revised version of the manuscript that addresses all the issues raised.

---

### Decision · Action_Editor_nowH · 2024-12-02

**Recommendation:** Reject

**Comment:**

The presentation of the method has several issues:

- The description of the approach is often unclear. It would help a lot to clearly define the variables K (number of measurements), N (number of image reconstructions using the current belief over the latents), and E (which is referred to as the number of "experiments", which is still a bit unclear to me). It would also be important to show the actual loss that is used during training (VAE + additional KL loss for partial encoder).

- Refering to Eq.9 as log-likelihood is wrong, as it corresponds to the difference between two likelihoods. Instead, it relates to a one-sample estimate of the KL between the posterior belief and prior belief (which is indeed a sensible criterium).

- Eq. 1 and Eq. 6 imply that $\mathbb{E}\_z \Big[ \log p(\mathbf{x}) \Big] = \log p(\mathbf{x}) \le 0$. However, this inequality is not satisfied in the continuous setting. Similarly, Eq.5 implies that $\mathbb{E}\_z \Big[ \log p(\mathbf{y^{[B^k]}}) \Big] = \log p(\mathbf{y^{[B^k]}}) \le 0$ which is not valid for continuous observation space. These issues do not affect the soundness of the approach (because the omitted terms can be dropped as they are constant w.r.t. the optimization variables).

- Minor: Wrong parantheses in Eq. 14, Missing subscript for $z_i$ in Eq. 10.


For these reasons, I argue that in its current state, the submission needs to be rejected. However, I encourage the authors to resubmit after significantly improving the presentation, and adding an experiment to better motivate application.

**Audience:**

While the approach may appeal to some of TMLR's readers, the submission's current relevance is limited due to the lack of a more realistic evaluation, especially given its modest novelty.

**Claims And Evidence:**

The submission introduces a novel method for active sequential inference, which involves selecting measurements based on a Variational Autoencoder (VAE) with a partial encoder. This VAE is used to sample possible reconstructions and evaluate the usefulness of different potential next measurement actions. The approach is evaluated on FashionMNIST and compared to stochastic variational inference.

Reviewer LXw8 stated in their review that the claims are not sufficiently supported, but did not specify which claims were lacking support and did not respond to follow-up questions and emails, nor did they submit an official recommendation. Their biggest complaint was that the method was only tried out on a toy problem, not a real active measurement situation.
While Reviewer JGk7 and FJyZ both stated that the claims and evidence are sufficiently supported, both reviewers noted the lack of realistic experimental evaluation, causing Reviewer FJyZ to lean towards rejection.

I agree with the general consensus that the main weakness of the submission is the lack of evaluation on a realistic use case. Furthermore, I argue that the work is not presented sufficiently well (see comments below).
In summary, the claims made in the submission are currently not supported by accurate, convincing and clear evidence, because several equations are wrong and the proposed method is not tested on an actual use case.

**Resubmission Of Major Revision:**

The authors may consider submitting a major revision at a later time.